# Euedaphic Rather than Hemiedaphic or Epedaphic Collembola Are More Sensitive to Different Climate Conditions in the Black Soil Region of Northeast China

**DOI:** 10.3390/insects16030275

**Published:** 2025-03-05

**Authors:** Chunbo Li, Shaoqing Zhang, Baifeng Wang, Zihan Ai, Sha Zhang, Yongbo Shao, Jing Du, Chenxu Wang, Sidra Wajid, Donghui Wu, Liang Chang

**Affiliations:** 1College of Life Science, Shenyang Normal University, Shenyang 110034, China15304434795@163.com (Y.S.); 2State Key Laboratory of Black Soil Conservation and Utilization, Northeast Institute of Geography and Agroecology, Chinese Academy of Sciences, Changchun 130102, China; zhangshaoqing@iga.ac.cn (S.Z.); aizihan@22mails.ucas.ac.cn (Z.A.); 18803220826@163.com (S.Z.); wangchenxu@iga.ac.cn (C.W.); wuconchuicaa@ac.cn (D.W.); 3Key Laboratory of Wetland Ecology and Environment, Northeast Institute of Geography and Agroecology, Chinese Academy of Sciences, Changchun 130012, China; pakistan.xidi888@nenu.edu.cn; 4Agro-Biotechnology Research Institute, Jilin Academy of Agriculture Sciences, Northeast Agricultural Research Center of China, Changchun 130033, China; wbfwanabaifena@163.com; 5Department of Zoology, University of Jhang, Jhang 35200, Pakistan; 6School of Environment, Northeast Normal University, Changchun 130024, China

**Keywords:** climate difference, land uses, springtail, life form, black soil region

## Abstract

Soil biodiversity is profoundly affected by variations in climate conditions and land use practices in the black soil region of Northeast China. While most studies focus on aboveground biodiversity, less is known about soil biodiversity. This study examined how climate and land use practices affect Collembola (springtails), a type of soil organism, in the black soil of Northeast China. Researchers sampled three climatic areas (from high to low latitudes) and three land use types (soybean, maize, and rice) in each area. They found that warmer, more humid climates and land use practices shifting from rice to soybean and maize increased Collembola density and species richness. Specifically, euedaphic Collembola (living deeper in soil) were more sensitive to climate differences, while all Collembola life forms responded positively to soybean and maize fields. Environmental factors and soil microorganisms significantly influenced Collembola communities, with environmental factors having stronger impacts. These findings suggest that the variations in climate conditions and land use types may alter the vertical distribution of soil fauna and affect related ecological processes in agricultural systems. This study highlights the importance of protecting soil biodiversity in the face of global environmental changes.

## 1. Introduction

Aboveground and belowground biodiversity is strongly influenced by climate conditions and land use types [1,2,3]. Temperature and humidity variations, as two representative factors reflecting climate conditions, are key factors threatening global species diversity [4,5]. Kardol et al. discovered a positive correlation between the soil organism diversity and soil water content, as well as a negative correlation with the soil temperature (a decrease in water content due to warming) [6]. Consequently, both warming and drought can have detrimental effects on soil biodiversity [7]. Furthermore, different land use types pose a broader threat to a variety of species [8], as alterations in land use can lead to modifications in the physical and chemical environment of the soil where soil organisms reside, ultimately impacting soil organisms themselves [9].

Collembola, some of the most abundant invertebrates, are recognized as a microphagous group that is regulated by feeding resources and environmental factors [10]. Most Collembola feed on a wide range of microorganisms and organic sources [11], and, through their trophic interactions, regulate the microbial community structure and activity, thus impacting the decomposition rate and nutrient cycling [12,13,14]. A wide variety of ecological and environmental factors affect Collembola communities [15], particularly modifications in soil chemical properties [16]; microclimatic conditions and microhabitat configurations [17,18]; as well as land use type and management practices [19,20]. Due to their high sensitivity to environmental changes, Collembola are often used as indicators to assess environmental degradation and soil quality [21,22].

Additionally, Collembola are classified into three main life forms, reflecting their vertical stratification along the soil profile [23,24]. Different life forms of Collembola may respond differently to different climate conditions and land use types, potentially leading to variations in the Collembola community structure [25,26]. It is believed that soil species inhabiting the surface (i.e., hemiedaphic and epedaphic Collembola) exhibit greater tolerance toward warmer and drier conditions compared to those living in the subsoil and deep soil layers [20,27]. Deep soil organisms, on the other hand, experience less abiotic environmental variability and may therefore be more susceptible to the temperature and moisture variations induced by warming and precipitation [28,29,30]. Furthermore, according to Ponge et al., euedaphic Collembola species may be more susceptible to different land use types due to their limited dispersal activity compared to hemiedaphic and epedaphic Collembola species [31]. The composition of Collembola functional groups in cropland is insufficiently studied, although it is known that grasslands and arable lands are more favored by fast-dispersing epedaphic species [31,32]. Collembolans respond differently to varying climate conditions and land use practices. Therefore, the interaction between these two factors may have a more profound impact on the composition and community structure of different Collembola life forms [33,34].

The black soil region of Northeast China is one of the three major black soil regions in the Northern Hemisphere and the largest commercial grain production base in China [35], with the crop yields of maize, rice, and soybean accounting for about 30% of the national total [36,37]. Over the years, due to excessive reclamation and utilization, the black soil area of Northeast China has gradually evolved from a natural ecosystem of forests and grasslands to an artificial farmland ecosystem. Consequently, the extent of grain cultivation has progressively expanded, with a particular emphasis on soybeans, maize, and rice, which collectively represent 98.9% of the total area dedicated to grain production [38]. Concurrently, the climate conditions in Northeast China are gradually changing and affecting its black soil agroecosystem. Over the past 50 years, the rate of increase in the regional average temperature has been 0.31 °C/10a, higher than the national average [39]. Future projections indicate that the average temperature in the Northeast black soil region is expected to rise [40], while humidity generally shows a fluctuating increase, but with increased heterogeneity [41]. These changes have not only affected the soil ecosystem functions in the black soil region of the Northeast [42,43] but also have an impact on the biodiversity of the region [44,45]. However, most of the previous studies have focused on aboveground biodiversity, while fewer studies have been conducted on belowground biodiversity [46]. In light of a clear lack of research on the subject, this study investigates the differences in Collembola communities under varying climate conditions and land use practices in the black soil region of Northeast China.

Firstly, the differences in climate conditions and land use practices result in different soil environments and feeding resources, such as temperature, humidity and soil microorganisms [47,48,49]. Therefore, we hypothesize (H1) that the Collembola density and species richness respond differently to different climate conditions and land use types.

Secondly, in contrast to hemiedaphic and epedaphic forms, euedaphic Collembola demonstrate substantially restricted mobility patterns [31], and we hypothesize (H2) that euedaphic Collembola are more sensitive to different climate conditions and land use practices.

Thirdly, feeding resources such as soil organic matter (SOM) are relatively high in the black soil region and this could not be a limiting factors for Collembola community [50]; so, we hypothesize (H3) that the soil environmental conditions (e.g., whether it is flooded or not, temperature, etc.) will have a greater influence on Collembola than the feeding resource (microorganisms).

## 2. Materials and Methods

### 2.1. Study Area

The study area was selected in the black soil region of the Northeast, located in the Sanjiang Plain area of Heilongjiang Province, China. The soils in this area are classified as black soils (Typical Hapludoll, USDA Soil Classification). We selected three different climatic areas to account for varying climatic conditions:(1)Fujin County (CK) is located in the northeast of Heilongjiang Province, on the south bank of the lower reaches of the Songhua River, and the soil type is typical chernozem. Our sampling site is located between latitudes 47.0847°–47.3745° N and longitudes 132.5501°–132.7848° E. The mean annual temperature (MAT) between the sampling sites is 2.61 °C, and the mean annual precipitation (MAP) is 556 mm (detailed sample site information is listed in Table A1).(2)Huanan County (with a higher temperature and higher humidity than Fujin) is located in the eastern part of Heilongjiang Province, at the foot of Wanda Mountain, a remnant of the Changbai Mountains, and the soil type is typical chernozem. Our sampling site is located between latitudes 46.2357°–46.3396° N and longitudes 129.9683°–130.5665° E, with a MAT of 3.23 °C and MAP of 567 mm between sampling sites. Compared with Fujin, Huanan has a higher temperature and higher humidity, with a MAT increase of 0.62 °C and a MAP increase of 11 mm.(3)Youyi County (with a higher temperature and lower humidity than Fujin) belongs to Shuangyashan City, Heilongjiang Province, is located in the northeastern part of Heilongjiang Province, the soil type is typical chernozem, and our sampling site is located between latitudes 46.7425°–46.9028° N and longitudes 131.4236°–131.9535° E with a MAT of 3.58 °C and MAP of 532 mm between sampling sites. Compared with Fujin, Youyi has a higher temperature and lower humidity, with a MAT increase of 0.97 °C and a MAP decrease of 23 mm.

By selecting these three distinct climatic areas, we aimed to capture the variations in the climate conditions present in the black soil area of Northeast China.

### 2.2. Experimental Design and Soil Sampling

In each region, we selected three types of arable land (soybean, maize, and rice). To minimize the impact of farming practices on the experiment, we identified conventional arable land with slopes < 2° and mechanized traditional cultivation for >10 years by consulting elderly residents of local villages. Fertilizer application rates in the Northeast black soil region are consistent, with 1050–1200 kg/ha of compound fertilizer applied to soybeans, 750–1050 kg/ha of compound fertilizer applied to maize, and 900–1050 kg/ha of compound fertilizer applied to rice; pesticide use is also consistent, with the use of herbicides (ethopropylamine and atrazine) (1500–3000 mL/ha) and insecticides (750–1500 mL/ha) in the spring sowing and summer crop growth periods. Due to these constraints, we selected 10 soybean, 9 maize, and 32 rice sample plots in Fujin County; 14 soybean, 27 maize, and 12 rice sample plots in Huainan County; and 10 soybean, 20 maize, and 17 rice sample plots in Youyi County. The geographic coordinates (latitude, longitude) of each site were recorded using a handheld GPS device (Venture; Garmin, Olathe, KS, USA).

Samples were collected in October 2023 (autumn), when the crops had matured and water had been discharged from the paddies. We used a 5.5 cm diameter soil auger to collect two soil samples at a 0–10 cm depth in the tillage layer, which were used to extract soil Collembola and soil microorganisms. A soil sample was immediately sent to the laboratory where Collembola were extracted from the soil sample using the Tullgren funnel method. The extracted Collembola were preserved in 75% ethanol, identified to the species level (Table A2), and classified into three life forms: euedaphic, hemiedaphic, and epedaphic Collembola [51,52,53,54,55]. The relative abundance of each Collembola species was calculated. Additionally, another sample was freeze-dried and stored at −80 °C for the PLFA analysis.

### 2.3. Climate and Soil Factors

Climate data were downloaded from WorldClim (https://worldclim.org/ (accessed on 5 November 2023)) and extracted as raster files of climate data in ArcGis using the band function, after which the ‘agda1’, ‘sp’, and ‘raster’ packages [56,57] were utilized in R4.4.2 to extract climate data for each sample point based on the latitude and longitude of the actual sampling point (Table A1). Soil pH was determined using a pH meter in 1:5 soil/water solution (*w*/*v*) (Table A3). Soil total carbon was determined by the volumetric method using potassium dichromate, and soil total nitrogen by the semi-micro Kjeldahl distillation method [58].

### 2.4. PLFA Analysis

The soil microbial community was characterized using a phospholipid fatty acid (PLFA) analysis, as described by Bossio and Scow [59]. Briefly, a chloroform–methanol–citrate buffer solution (1:2:0.8, *v*/*v*/*v*) was used to extract lipids from 8 g of freeze-dried soil. After extraction, the nonpolar lipids were fractionated into phospholipids by solid-phase extraction columns (Supelco Inc., Bellefonte, PA, USA) and transformed into fatty acid methyl esters using mild alkaline methanolysis. Fatty acid 19:0 was used as internal standard and added to fatty acid methyl esters. After this addition, the samples were analyzed and identified by an Agilent 6850 series Gas Chromatograph (Agilent, made in Shanghai, China) with MIDI version 6.0 peak identification software.

The following biomarkers were used: total PLFAs (from C14 to C20), Gram-positive bacterial (G+) PLFAs (i-14:0, i-15:1ω6c, i-15:0, a-15:0, i-16:0, i-17:1ω9c, i-17:0, and a-17:0), Gram-negative bacterial (G−) PLFAs (16:1ω9c, 16:1ω7c, 17:1ω8c, cy-17:0ω7c, 18:1ω7c, 18:1ω6c, 18:1ω5c, cy-19:0ω9c, cy-19:0ω7c, and 20:1ω9c), anaerobe PLFAs (14:0, 16:0, 18:0, i-15:0, a-15:0, a-15:0, i-17:0, a-17:0, cy-17:0, cy-19:0, 16:1ω7c, and 18:1ω7c), fungal PLFAs (18:3ω6c, 18:2ω6c, 18:1ω9c), arbuscular mycorrhizal fungal (AMF) PLFAs (16:1ω5c), eukaryotic PLFAs (18:2ω6,9, 18:3ω3,6,9, 20:4ω6, 20:5ω3, 22:6ω3, 18:1ω9c, 16:1ω5c, 16:0 2OH, 18:0 2OH, 17:1ω8c, and 19:1ω8c) and actinomycete PLFAs (10Me-C16:0, 10Me-C17:1ω7c, 10Me-C17:0, 10Me-C18:1ω7c, 10Me-C18:0, and 10Me-C20:0) [60,61].

### 2.5. Statistical Analysis

The residual normality and variance heterogeneity of all response variables were tested using the ‘check_normality’ function and ‘check_homogeneity’ function of the ‘performance’ package [62] before data analysis. After confirming that all response variables did not conform to residual normality, they conformed to a normal distribution with variance heterogeneity after the log(x + 1) transformation of all response variables. Therefore, we used two-way ANOVA to analyze differences in all response variables under varying climate conditions and land use practices. When the ANOVAs indicated significant treatment effects, post hoc Tukey’s HSD tests were conducted to test for differences among the respective levels within factors (Table A4). All data analyses were performed in R4.4.2.

To explore the relationships between the Collembola community composition, land use practices, climate conditions, and the measured PLFA values of the microorganisms, a constrained redundancy analysis (RDA) was performed using the ‘vegan’ package with permutation tests (permutation number: 999) [63,64,65]. We checked for linear relationships in the datasets (Euclidean metric; prerequisite for this method) by performing detrended correspondence analyses (DCA) and identifying the respective longest gradient. As these were always lower than 3, the use of linear methods was considered appropriate [66]. Subsequently, the ‘envift’ test was performed using the ‘envift’ function of the ‘vegan’ package as a means of determining the effect of each factor on the composition of the Collembola community [65].

## 3. Results

### 3.1. Effects of Differences in Climate Conditions and Land Use Practices on Microorganisms

There were significant differences in AMF, G−, anaerobe, and actinomycete, while there were no significant differences in eukaryote, fungi, and G+ under different climate conditions (Table 1). On the other hand, different land use practices had significant effects on AMF, G−, G+ and actinomycetes (Table 1). Furthermore, varying climate conditions and land use practices only had significant interaction effects on AMF, with marginal effects on fungi and no significant effects on other microorganisms (Table 1).

### 3.2. Effects of Differences in Climate Conditions and Land Use Practices on the Total Collembola Density and Species Richness

Different climate conditions had significant effects on both the density and species richness of the total Collembola community (Table 2, Figure 1). In contrast, different land use practices had significant effects on both the density and species richness of the total Collembola community. On the one hand, the density and species richness of the total Collembola community within soybean and maize fields were much higher than that of rice fields. On the other hand, at a higher temperature and higher humidity, the total Collembola density and species richness were much higher; while at a higher temperature and lower humidity, the total Collembola density and species richness did not differ significantly (Figure 1). Furthermore, no interaction was found between different climates and land use practices concerning the total density and species richness of Collembola (Table 2, Figure 1).

### 3.3. Effects of Differences in Climate Conditions and Land Use Practices on the Density and Species Richness of Epedaphic, Hemidaphic, and Euedaphic Collembola

Different life forms of Collembola responded differently to varying climate conditions, with different climate conditions having a significant effect on the density and species richness (Table 3, Figure 2) of euedaphic but not epedaphic and hemiedaphic Collembola (Table 3, Figure 3 and Figure 4). As for different land use practices, they had significant effects on the density and species richness of all three life forms of Collembola (Table 3, Figure 2, Figure 3 and Figure 4). Furthermore, euedaphic Collembola had a higher density and species richness at a higher temperature and higher humidity; while all three life forms Collembola had a higher density and species richness in soybean and maize fields compared to rice fields. Interestingly, an interaction between climate and land use practices was observed for the species richness of hemiedaphic Collembola, although it did not affect other life forms of Collembola (Table 3, Figure 3).

### 3.4. Effects of Environmental and Feeding Resources on Collembola Communities

Land use practices, MAP, fungi, AMF and G− significantly affected the variation in the Collembola community composition and they explained 40.1%, 6.1%, 4.8%, 4.6%, and 4.1% of the total variance, respectively (Table 4).

A total of 20.85% of the variance in the data was accounted for by 10 RDA axes, where the first RDA axis explained 72.46% of the variance (Table 5), with land use practices being the main explanatory factor, and the highest abundance of *Thalassaphorura macrospinata* species was observed (Figure 5). The second RDA axis accounted for 16.7% of the variance, with MAP being the most influential factor, and the highest presence of *Proisotoma minuta* species was recorded.

## 4. Discussion

### 4.1. Effects of Differences in Climate Conditions and Land Use Practices on Collembola Communities

Our study demonstrates that different climate conditions have a slight effect on Collembola density but a substantial impact on the Collembola species richness, which mostly supports our hypothesis (H1) that both the Collembola density and richness respond differently under varying climate conditions. This finding is further supported by the studies conducted by Li et al. and Makkonen et al. [28,67]. Moreover, we observe that higher humidity and temperature have strong influences on Collembola. This could be attributed to the fact that humidity plays a crucial role in shaping Collembola communities [6]. Additionally, Thakur et al. demonstrated that high temperature negatively impacts the feeding behavior of soil trophic fauna only when humidity levels decrease, as also supported by our RDA results [68]. Conversely, under suitable conditions where resources are abundant and humidity is optimal, high temperature can positively affect Collembola populations [69,70]. Previous studies in the same research area have shown that increased temperatures can enhance both Collembola density and species richness [71].

On the other hand, our results show that soybean and maize fields have higher Collembola densities and species richness than paddy fields, which largely supports our hypothesis (H1) that different land use practices significantly affect the Collembola density and species richness. Saifutdinov et al. demonstrated that rice cultivation negatively impacts Collembola communities [72]. One major reason for this observation is that paddy fields harbor limited soil fauna due to factors such as waterlogging, anaerobiosis, low temperatures, and a relative scarcity of microorganisms [73]. Additionally, the practice of rice cultivation could increase soil compaction [74]; this could negatively affect the soil Collembola community. These findings suggest that the deteriorated soil conditions under rice cultivation contribute to the reductions in Collembola density and species richness.

In contrast to our hypothesis (H1), we found no effect of the interaction between variations in climate conditions and land use practices on Collembola communities. The observed independent and significant effects of both variations in climate and land use practices on Collembola communities in our study may be attributed to the predominant role of land use practices as a key determinant of soil biodiversity [75,76]. Specifically, land use practices, particularly the differences between soybean and maize fields and paddy fields, have profound effects on the soil environment, altering the soil microclimate and influencing Collembola communities [9,77]. While different climate conditions do affect Collembola communities, their impacts do not mitigate the effects of differences in land use practices. Consequently, it can be inferred that the influence of differences in land use practices on Collembola communities remains unaffected by varying climate conditions.

### 4.2. Effectss of Differences in Climate Conditions and Land Use Practices on Epedaphic, Hemidaphic, and Euedaphic Collembola Communities

Our study demonstrates that different climate conditions have significant effects on euedaphic Collembola but not on epedaphic and hemiedaphic Collembola, which exactly supports our hypothesis (H2). Epedaphic Collembola exhibit a greater dispersal capacity [78], enabling them to escape horizontally rather than spreading vertically into deeper soil layers, as commonly observed in hemiedaphic and epedaphic Collembola, thus allowing them to locate more favorable microhabitats under adverse conditions [20]. Consistent with Ferrín et al. and Holmstrup and Bayley, epedaphic Collembola can more easily evade unsuitable abiotic conditions [79,80]. Additionally, soil-dwelling Collembola often encounter challenges related to food scarcity, which contrasts with the higher abundance present in upper soil layers.

Contrary to our hypothesis (H2), we find that different land use practices between rice fields and soybean and maize fields significantly affect not only euedaphic Collembola but also hemiedaphic and epedaphic Collembola. This observation may be attributed to the timing of our sampling, which coincided with the maturity phase of the rice. Despite the drainage of water from the rice fields, epedaphic and hemiedaphic Collembola had not sufficiently adapted to alterations in the soil environment characteristics of the flooded rice field. Consequently, they were unable to rapidly recolonize the soil following the cessation of the intensive disturbance. Alternatively, the negative effects of the biotope type on the abundance of euedaphic Collembola underlines the sensitivity of these groups to soil degradation in rice-based agroecosystems. Euedaphic Collembola rely heavily on the integrity of the soil pore microstructure and are particularly affected by soil compaction, which tends to be higher in rice-based cropping systems due to periodic ponding and flooding [81], and euedaphic Collembola are more susceptible [82].

Furthermore, it is observed that the interaction between varying climate conditions and land use practices only affects the species richness of hemiedaphic Collembola, without impacting euedaphic and other hemiedaphic Collembola. Our study reveals that higher MAP and MAT result in a higher species richness of hemiedaphic Collembola. This can be attributed to the fact that higher levels of precipitation and temperature are favorable for hemiedaphic Collembola. Additionally, these differences may enhance the adaptation of some hemiedaphic Collembola to the effects of land use practices.

### 4.3. Effects of Environmental and Feeding Resources on Collembola Communities

Our study demonstrates that both environmental factors and feeding resources significantly affect Collembola communities, and environmental factors have a greater effect on Collembola communities (land use practices and MAP together explained 46.2% of the total variance respectively (Table 4)), whereas feeding resources have a lesser effect (fungi and AMF together explained 9.4%), which directly supports our hypothesis (H3). Land use practices affect biodiversity by altering natural habitats [83]. For instance, varying land use practices between drylands and paddy fields drastically modify the habitat of Collembola, particularly the soil porosity [73], reducing their survival space. In addition, MAP explained 6.1% of the total variance, which is relatively comparable to the contributions of fungi (4.8%) and AMF (4.6%) for feeding resources. Therefore, the environmental difference caused by varying land use practices has a much greater impact on Collembola communities than other factors. The influence of fungal-dominated feeding resources on Collembola can be attributed to factors such as increased precipitation and temperature leading to elevated metabolic rates and heightened biological activity in Collembola [84,85]. Consequently, biological processes in Collembola, including feeding and movement, tend to accelerate [86,87], compelling Collembola to rely more heavily on feeding resources such as AMF to meet their energy demands.

## 5. Conclusions

The results of this study indicate that drylands could harbor more soil fauna species for biodiversity conservation in agricultural land, since Collembola levels were sharply decreased in rice fields. Moreover, environmental factors were the dominant limiting conditions to Collembola community alterations rather than feeding resources in the black soil region of Northeast China. Higher temperature and higher humidity climate favored euedaphic Collembola rather than hemiedaphic and epedaphic groups. This indicated that climate differences could alter the vertical distribution characteristics of soil fauna (e.g., increasing soil-dwelling fauna) as well as alter the ecological processes associated with soil fauna under future global changes.

## Figures and Tables

**Figure 1 insects-16-00275-f001:**
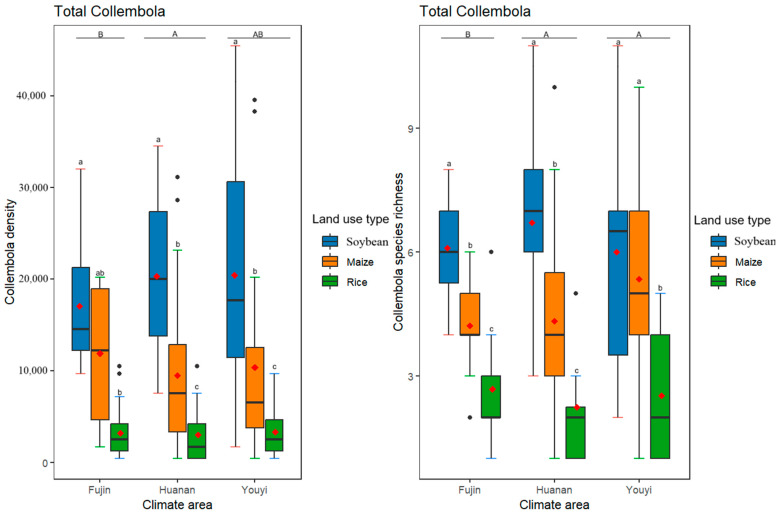
Effects of differences in climate conditions and land use practices on the total Collembola density and species richness. The box plots display the medians (horizontal lines), means (red diamonds), first and third quartiles (rectangles) and outliers (isolated dots). Uppercase letters represent multiple comparisons of climate conditions and lowercase letters represent multiple comparisons of land use practices, different letters above the bars indicate significant differences among treatments (*p* < 0.05), two letters represent marginal differences (0.05 < *p* < 0.1). Significance was determined as *p* < 0.05 by post hoc Tukey’s HSD tests. The horizontal axis of the graph represents various climatic regions.

**Figure 2 insects-16-00275-f002:**
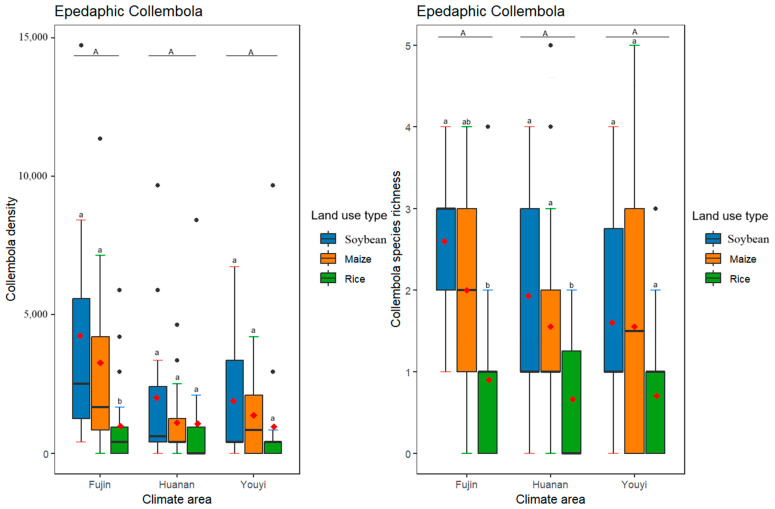
Effects of differences in climate conditions and land use practices on epedaphic Collembola density and species richness. Box plots show the medians (horizontal lines), means (red diamonds), first and third quartiles (rectangles) and outliers (isolated dots). Uppercase letters represent multiple comparisons of climate conditions and lowercase letters represent multiple comparisons of land use practices, different letters above the bars indicate significant differences among treatments (*p* < 0.05), two letters represent marginal differences (0.05 < *p* < 0.1). *p* < 0.05 by post hoc Tukey’s HSD tests. The horizontal axis of the graph represents various climatic regions.

**Figure 3 insects-16-00275-f003:**
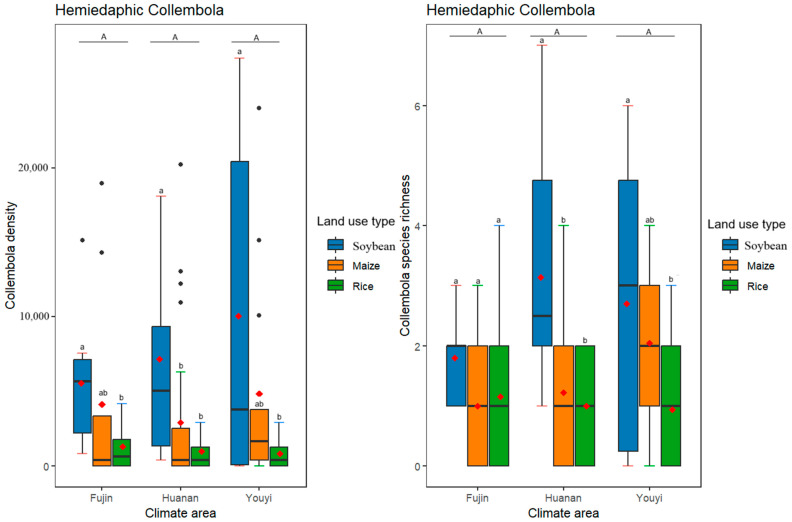
Effects of differences in climate conditions and land use practices on the density and species richness of hemiedaphic Collembola. Box plots show the medians (horizontal lines), means (red diamonds), first and third quartiles (rectangles) and outliers (isolated dots). Uppercase letters represent multiple comparisons of climate conditions and lowercase letters represent multiple comparisons of land use practices, different letters above the bars indicate significant differences among treatments (*p* < 0.05), two letters represent marginal differences (0.05 < *p* < 0.1). *p* < 0.05 by post hoc Tukey’s HSD tests. The horizontal axis of the graph represents various climatic regions.

**Figure 4 insects-16-00275-f004:**
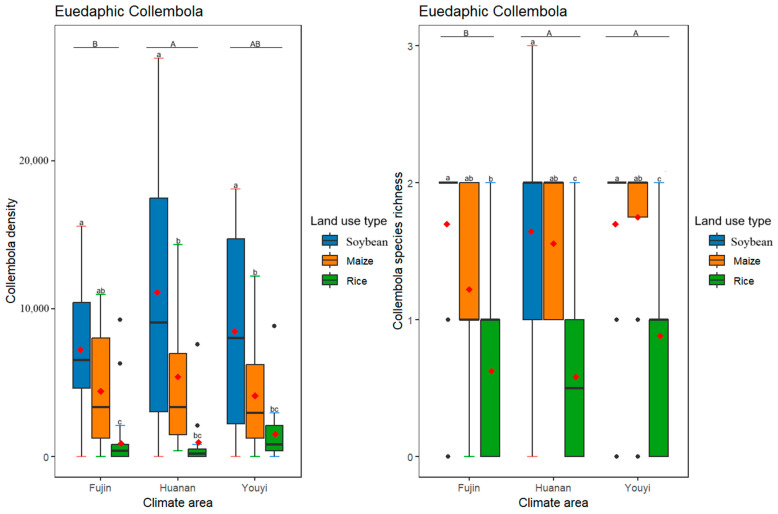
Effects of differences in climate conditions and land use practices on the density and species richness of euedaphic Collembola. Box plots show the medians (horizontal lines), means (red diamonds), first and third quartiles (rectangles) and outliers (isolated dots). Uppercase letters represent multiple comparisons of climate conditions and lowercase letters represent multiple comparisons of land use practices, different letters above the bars indicate significant differences among treatments (*p* < 0.05), two letters represent marginal differences (0.05 < *p* < 0.1). *p* < 0.05 by post hoc Tukey’s HSD tests. The horizontal axis of the graph represents various climatic regions.

**Figure 5 insects-16-00275-f005:**
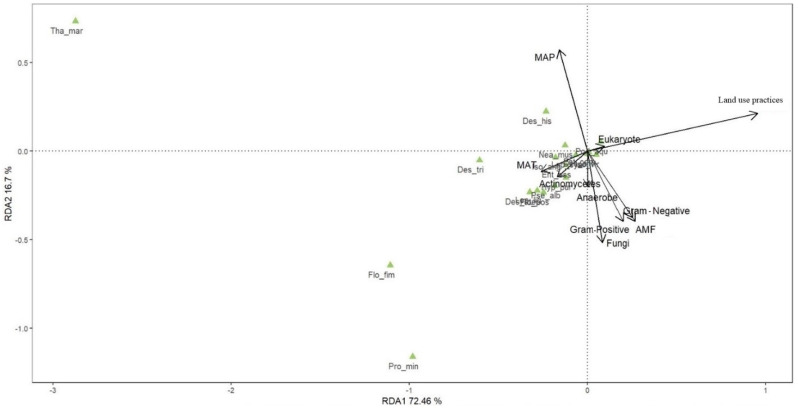
Species treatment plot based on the redundancy analysis (RDA) of the Collembola community composition. About 20.85% of the total variance in the dataset was explained by the 10 constrained RDA axes. Of these, RDA axes 1 and 2 explained 72.46% and 16.7% of the variance, respectively. These include climate (MAT and MAP), land use practices, and microorganisms (actinomycetes, AMF, anaerobe, eukaryote, fungi, G−, G+). For Collembola species: *Pro_min* = *Proisotoma minuta*, *Pse_alb* = *Pseudosinella alba*, *Lep_lig* = *Lepidocyrtus lignorum*, *Iso_vir* = *Isotoma viridis*, *Des_ate* = *Desoria ater*, *Flo_fim* = *Folsomia fimetaria*, *Tha_mar* = *Thalassaphorura macrospinata*, *Flo_pos* = *Folsomia postsensilis*, *Ent_com* = *Entomobrya comparata*, *Lep_cya* = *Lepidocyrtus cyaneus*, *Ent_ass* = *Entomobrya assuta*, *Nea_mus* = *Neanura muscorum*, *Hyp_pur* = *Hypogastrura purpurescens*, *Des_tri* = *Desoria tigrina*, *Pod_aqu* = *Podura aquatica*, *Iso_ant* = *Isotomurus antennalis*, *Iso_ang* = *Isotoma anglicana*, and *Des_his* = *Desoria hissarica*.

**Table 1 insects-16-00275-t001:** The F-values and *p*-values of the ANOVA results are presented in Table 1. This study examined the effects of differences in climate conditions and land use practices and their interactions on AM fungi, G−, G+, eukaryotes, fungi, anaerobes, and actinomycetes. The significance levels were defined as follows: (*) represents 0.05 < *p* < 0.1, * represents *p* < 0.05, ** represents *p* < 0.01, and *** represents *p* < 0.001.

Effects	Df		Microorganism						
			Actinomycete	Anaerobe	AMF	Eukaryote	Fungus	G+	G−
Climate (C)	2	F	15.937	5.048	10.789	0.647	1.22	1.920	11.754
		*p*	<0.001 ***	0.008 **	<0.001 ***	0.525	0.295	0.150	<0.001 ***
Land use (L)	2	F	5.397	1.873	5.035	0.262	0.425	3.316	8.068
		*p*	0.005 **	0.157	0.007 **	0.770	0.654	0.039 *	<0.001 ***
C × L	4	F	0.414	0.874	5.906	0.137	2.026	1.444	1.728
		*p*	0.798	0.481	<0.001 ***	0.968	0.0939 (*)	0.223	0.147

**Table 2 insects-16-00275-t002:** The F-values and *p*-values obtained from ANOVA. This study investigated the effects of differences in climate conditions and land use practices, as well as their interactions, on both the total Collembola density and species richness.

Effects	Df		Total Collembola	
			Collembola Density	Collembola Species Richness
Climate (C)	2	F	2.812	3.425
		*p*	0.063 (*)	0.035 *
Land use (L)	2	F	47.348	43.132
		*p*	<0.001 ***	<0.001 ***
C × L	4	F	0.480	1.136
		*p*	0.751	0.342

The significance levels are indicated as follows: (*) represents 0.05 < *p* < 0.1, * represents *p* < 0.05, and *** signifies *p* < 0.001.

**Table 3 insects-16-00275-t003:** The F-values and *p*-values of the ANOVA results. Effects of differences in climate conditions and land use practices and their interactions on the density and species richness of epedaphic, hemiedaphic, and euedaphic Collembola.

**Effects**	**Df**		**Epedaphic Collembola**	
			**Collembola Density**	**Collembola Species Richness**
Climate (C)	2	F	1.482	0.418
		*p*	0.231	0.659
Land use (L)	2	F	7.125	15.364
		*p*	0.002 **	<0.001 ***
C × L	4	F	1.700	0.417
		*p*	0.153	0.796
**Effects**	**Df**		**Hemiedaphic Collembola**	
			**Collembola Density**	**Collembola Species Richness**
Climate (C)	2	F	1.331	2.320
		*p*	0.267	0.102
Land use (L)	2	F	13.852	14.529
		*p*	<0.001 ***	<0.001 ***
C × L	4	F	0.751	2.544
		*p*	0.559	0.042 *
**Effects**	**Df**		**Euedaphic Collembola**	
			**Collembola Density**	**Collembola Species Richness**
Climate (C)	2	F	5.630	7.724
		*p*	0.003 **	<0.001 ***
Land use (L)	2	F	27.401	28.979
		*p*	<0.001 ***	<0.001 ***
C × L	4	F	0.667	0.552
		*p*	0.616	0.698

The significance levels were defined as follows: * represents *p* < 0.05, ** represents *p* < 0.01, and *** represents *p* < 0.001.

**Table 4 insects-16-00275-t004:** Envift tests for land use practices, MAP, MAT, and microorganisms, where the bold font indicates a significant effect.

Effect	R^2^	*p*-Value
Land use practices	**0.401**	**0.001**
MAP	**0.061**	**0.007**
Fungi	**0.048**	**0.023**
AMF	**0.046**	**0.024**
G−	**0.041**	**0.033**
G+	0.036	0.058
MAT	0.033	0.084
Actinomycetes	0.018	0.207
Anaerobe	0.007	0.553
Eukaryote	0.004	0.714

**Table 5 insects-16-00275-t005:** Permutation tests for each axis of the RDA.

Axis	Df	Var	F	*p*
RDA1	1	43.02086	26.71935	0.001
RDA2	1	9.913363	6.15698	0.128
RDA3	1	2.61313	1.62296	0.988
RDA4	1	1.335855	0.829672	1
RDA5	1	1.078449	0.669802	1
RDA6	1	0.722842	0.448942	1
RDA7	1	0.377377	0.234381	1
RDA8	1	0.153134	0.095108	1
RDA9	1	0.109407	0.06795	1
RDA10	1	0.047508	0.029506	1

## Data Availability

The data are available upon request.

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
