# Peer review of "Euedaphic Rather than Hemiedaphic or Epedaphic Collembola Are More Sensitive to Different Climate Conditions in the Black Soil Region of Northeast China"

_insects, 2025, doi:10.3390/insects16030275_

Round 1
Reviewer 1 Report
Comments and Suggestions for Authors
Some minor corrections/suggestions in the attached pdf.
If you make the changes suggested in Table S2, please check that the results do not change.

Author Response
Reviewer: 1
Comments to the Author
Some minor corrections/suggestions in the attached pdf. If you make the changes suggested in Table S2, please check that the results do not change.
Response: Thank you for your valuable comments and suggestions on our research. We highly appreciate your feedback, which we have carefully considered and will incorporate into the revised manuscript. The results did not change after we corrected it according to your comments.
Introduction
Comments 1: lines 61. Add et al.
Response 1: Agree. we add “et al.”, please see line 61.
Comments 2: lines 77, - modified with ,.
Response 2: Agree, we change “-“ to “,” ,please see line 77.
Comments 3: lines 87. Add et al.
Response 3: Agree, we add “et al.”, please see line 87.
Comments 4: lines 301. Thalassaphorura macrospinata modified with Thalassaphorura macrospinata.
Response 4: Agree, we change “Thalassaphorura macrospinata” to “Thalassaphorura macrospinata”, please see line 301.
Comments 5: lines 303. Proisotoma minuta modified with Proisotoma minuta.
Response 6: Agree, we change “Proisotoma minuta” to “Proisotoma minuta”, please see line 303.
Comments 7: lines 317 and 318. Neaura muscorum modified with Neanura muscorum, Desoria antennalis modified with Isotomurus antennalis.
Response 7: Agree, we change “Neaura muscorum” to “Neanura muscorum”, “Desoria antennalis” to “Isotomurus antennalis”, please see 317 and 318.
Comments 8: lines 326, 329, 337. Add et al.
Response 8: Agree, we add “et al.”, please see lines 326, 329, 337.
Comments 9: lines 364. Add et al.
Response 9: Agree, we add “et al.”, please see line 364.
Comments 10: lines 364. Add and Bayley.
Response 10: Agree, we add “and Bayley”, please see line 364.
Comments 11: lines 440, modified Table S2.
Response 11: Agree, we modified Table S2, please see line 440.
Reviewer 2 Report
Comments and Suggestions for Authors
Line 20-22, Remove butï¼›
Line 24-26: Therefore, we selected three climatic areas from high to low latitudes in the black soil region of Northeast, with three different land uses (soybean, maize, and rice) sampled in each area. Revise to Here, we selected three climatic areas from high to low latitudes in the black soil region of Northeast, with three different land uses (soybean, maize, and rice) sampled in each area.
Line 26-28: we found that higher temperature and higher humidity(HTHH)climate changes, this sentence needs to be rephrased, because temperature and humidity are belongs to climate.
Line 28-30: Specifically, HTHH climate change significantly increased euedaphic Collembola density and species richness, while land use change from rice to soybean and maize significantly increased all three life forms Collembola density and species richness, this sentence has same means as the previous sentence Line 26-28, which need to modify it.
Line 44-47: The author wants to report temperature and precipitation effect biodiversity, but this paragraph indicated that the water and temperature of soil have significant effect on biodiversity, what are you want to indicated?
Line 52: the microphagous group, which is regulated by feeding resources and environmental factors, do you want to discuss the feeding resources?, in this article, the main title is to indicate the environment factors effect group, so to tell researchers that the environment factors have significant effect on this Group.
Line 64-68: Wouldn't it be better to separate precipitation and land use change, this paragraph says climate and land use change, then precipitation and humidity, then land use change, the author needs to reorganize the language.
Line80-84: please transport it in Line86-90, and authors need to reorganize their language to clarify the impact of climate and land intensification on their biodiversity.
Line 205: p modify to p, follows in full text.
Line 212: The f-value should F- value.
Line 224-225: this sentence was wrong, HTHH climate change?, HTLH climate change, should write full name.
Line 269, where is the Fig. 4?
Line 287, Table 5. ANOVA test for each axis of RDA send in the appendix.
Line 303: Add both before density and richness
Line 304: remove And
Line 307: a strong influence replaces with strongly influence
Line 310: evidenced modified to supported.
Line 312: favorable modified to optimal
Line 312: “effect” is a noun, effect can modify to affect.
Line 315: fileld was spelling mistake, fields.
Line 317: significant modified to significantly.
Line 327: remove first change
Line 330: paddyland? Means paddy land, the same as Line 373.
Line 342: Consist with modified with Consistent with
Line 354: soil deterioration modified with soil degradation.
Line 371: exactly supposed modified with directly supports
Line 375: which are relatively comparable to modified with which is relatively comparable to.
Author Response
Reviewer: 2
Thank you for your valuable comments and suggestions on our research. We highly appreciate your feedback, which we have carefully considered and will incorporate into the revised manuscript.
Comments 1: Line 20-22, Remove butï¼›
Response 1: Agree, we remove but, please see line 37.
Comments 2: Line 24-26: Therefore, we selected three climatic areas from high to low latitudes in the black soil region of Northeast, with three different land uses (soybean, maize, and rice) sampled in each area. Revise to Here, we selected three climatic areas from high to low latitudes in the black soil region of Northeast, with three different land uses (soybean, maize, and rice) sampled in each area.
Response 2: Agree, we change “we selected three climatic areas from high to low latitudes in the black soil region of Northeast, with three different land uses (soybean, maize, and rice) sampled in each area.” to “Here, we selected three climatic areas from high to low latitudes in the black soil region of Northeast, with three variations land use practices (soybean, maize, and rice) sampled in each area.” Please see lines 40-42.
Comments 3: Line 26-28: we found that higher temperature and higher humidity(HTHH)climate changes, this sentence needs to be rephrased, because temperature and humidity are belongs to climate.
Response 3: Agree, we delete the “climate changes”, please see lines 42.
Comments 4: Line 28-30: Specifically, HTHH climate change significantly increased euedaphic Collembola density and species richness, while land use change from rice to soybean and maize significantly increased all three life forms Collembola density and species richness, this sentence has same means as the previous sentence Line 26-28, which need to modify it.
Response 4: Agree, we change the sentence: Specifically, the density and species richness of eudaphic Colmbola are higher in climate conditions with higher temperatures and humidity, while all three life forms Collembola density and species richness are higher in land use practices from rice to soybean and maize. Please see lines 44-47.
Comments 5: Line 44-47: The author wants to report temperature and precipitation effect biodiversity, but this paragraph indicated that the water and temperature of soil have significant effect on biodiversity, what are you want to indicated?
Response 5: Agree, we replaced “precipitation” with “humidity” to ensure consistency in the meaning of the entire sentence and paragraph. Please see lines 59-60.
Comments 6: Line 52: the microphagous group, which is regulated by feeding resources and environmental factors, do you want to discuss the feeding resources?, in this article, the main title is to indicate the environment factors effect group, so to tell researchers that the environment factors have significant effect on this Group.
Response 6: Agree, thank you very much for this suggestion. We primarily considered the direct impacts of climate change and land use change, but we also wanted to further determine whether the environmental changes caused by climate change and land use change or the changes in food resources have a greater impact on collembolans. Since collembolans feed on microorganisms, we used the phospholipid fatty acid (PLFA) method to assess microbial characteristics as an indicator of food resource changes. Given that the black soil region is characterized by high organic matter content and abundant food resources, we believe that food resources may not be a limiting factor, and that environmental conditions have a greater impact on collembolan communities than food resources (see hypothesis 3, lines 122-126). We use environmental factors and microbial characteristics in RDA to determine whether environmental conditions or food resources have a greater influence on changes in collembolan communities (see Tables 4-5 and Figure 5).
Comments 7: Line 64-68: Wouldn't it be better to separate precipitation and land use change, this paragraph says climate and land use change, then precipitation and humidity, then land use change, the author needs to reorganize the language.
Response 7: Agree, thank you very much for this suggestion. This is a concluding sentence that begins by describing how different life forms of Collembola are affected differently. The response of different life forms of Collembola to climate is presented in lines 81-86; the response of different life forms of Collembola to land use is presented in lines 86-91.
Comments 8: Line80-84: please transport it in Line86-90, and authors need to reorganize their language to clarify the impact of climate and land intensification on their biodiversity.
Response 8: Agree, thank you very much for this suggestion. in lines 98-102 we describe past trends in land use in the black soil region of northeast; in lines 102-108 we describe trends in climate in the black soil region of northeast; and then we describe the impacts of these on biodiversity in lines 108-110.
Comments 9: Line 205: p modify to p, follows in full text.
Response 9: Agree, we delete the p-value from the text and keep the p-value from the table.
Comments 10: Line 212: The f-value should F- value.
Response 10: Agree, We change “f-value” to “F-value”, please see lines 231, 248 and 272.
Comments 11: Line 224-225: this sentence was wrong, HTHH climate change?, HTLH climate change, should write full name.
Response 11: Agree, we change the sentence: On the other hand, under higher temperature and higher humidity, the total Collembola density and species richness much higher; while under higher temperature and lower humidity, total Collembola density and species richness did not differ significantly. Please see lines 242-245.
Comments 12: Line 269, where is the Fig. 4?
Response 12: Agree, we rechecked the paper and added Figure 4.
Comments 13: Line 287, Table 5. ANOVA test for each axis of RDA send in the appendix.
Response 13: Agree, we added each RDA axis to table 5.
Comments 14: Line 303: Add both before density and richness.
Response 14: Agree, we add “both” before density and richness, please see lines 324.
Comments 15: Line 304: remove And.
Response 15: Agree, we remove “And”, please see lines 325.
Comments 16: Line 307: a strong influence replaces with strongly influence.
Response 16: Agree, we modified to “with strong influence”, please see line 327.
Comments 17: Line 310: evidenced modified to supported.
Response 17: Agree, we change “evidenced” to “upported”, please see line 331.
Comments 18: Line 312: favorable modified to optimal.
Response 18: Agree, we change “favorable” to “optimal”, please see line 332.
Comments 19: Line 312: “effect” is a noun, effect can modify to affect.
Response 19: Agree, we change “effect” to “affect”, please see line 332.
Comments 20: Line 315: fileld was spelling mistake, fields.
Response 20: Agree, we change “fileld” to “fields”, please see line 335.
Comments 21: Line 317: significant modified to significantly.
Response 21: Agree, we change “significant” to “significantly”, please see line 337.
Comments 22: Line 327: remove first change.
Response 22: Agree, we modified the sentence: The observed independent and significant effects of both variations in climate and land use practices on Collembola communities in our study may be attributed to the pre-dominant role of land use practices as a key determinant of soil biodiversity. Please see lines 346-349.
Comments 23: Line 330: paddyland? Means paddy land, the same as Line 373.
Response 23: Agree, we change “paddyland” to “paddy fields”, please see line 350 and 395.
Comments 24: Line 342: Consist with modified with Consistent with.
Response 24: Agree, we change “Consist with” to “Consistent with”, please see line 363.
Comments 25: Line 354: soil deterioration modified with soil degradation.
Response 25: Agree, we change “soil deterioration” to “soil degradation”, please see line 376.
Comments 26: Line 371: exactly supposed modified with directly supports.
Response 26: Agree, we change “exactly supposed” to “directly supports”, please see line 393.
Comments 27: Line 375: which are relatively comparable to modified with which is relatively comparable to.
Response 27: Agree, we change “which are relatively comparable to” to “which is relatively comparable to”, please see line 397.
Reviewer 3 Report
Comments and Suggestions for Authors
The manuscript compares the Collembola assemblages of three areas with three types of crop plant. The study has interesting and useful results, however, the authors do not interpret their own results properly. At first I thought they were just using the concepts incorrectly, but later I realized they were also interpreting them incorrectly. My main problem is this: climate change and land use change? The study did not investigate changes but rather differences! Change is the term, if something becomes different in time, but you investigated static areas with given climates ONCE. If you want to investigate changes, you should sample your objects several times and then you can see whether it will change or not. This is a serious problem, and without it being corrected, this manuscript cannot be published in my opinion.
Title:
This study does not investigate climate change. I suggest to reword the title. This is now misleading.
Abstract
Line 22: You compared three different areas with slightly different climates. What kind of change is it?
Line 26: “higher temperature and higher humidity (HTHH) climate change” This is climate difference. This is not change.
Line 27: “land use change from rice to soybean and maize” It would be a change, if the crop land would have been previously rice then it would have changed into soybean. But in your study it is not the case. This word change and also this sentence make no sense. This “land use change” cannot “increase” the density of Collembola. I tell you what you detected. You detected different density and diversity values between the land use practices. And the same problem is with “climate change” in the WHOLE MANUSCRIPT. You need to rethink what you tested, what results you got, and what it all means.
Introduction:
The Introduction is misleading. The reader thinks that the topic and the study will be about a real climate change and land-use change experiment. But all the investigation is not about such an experiment.
Line 81: forests and grasslands
Lines 95-108: I suggest to rewrite all the hypotheses because you did not investigate climate and land use change…
Materials and methods:
Line 115: what does CK stand for?
Lines 126, 133: It is very strange that you call „climate change” the geographical differences between climates. Climate change means that the climate changes IN TIME, not SPACE! I suggest not to call it climate change, maybe „difference in climate”.
A map about the study sites would tell more than Table S1.
Lines 135-136: Three “static” climatic area will be compared. Where is the climate change? Did these areas had lower temperatures and higher precipitations some years ago? Did you also investigate the Collembola some years ago on these areas? Then this study is not about climate change…
Lines 138-141: Three types of crop plants. Were these areas had previously different crop plants e.g. in 2022 or in 2021? Did you investigate the Collembola before on these areas? Then where is “land use CHANGE”?
Line 144: what are the amounts and timings of pesticide uses?
Line 190: Climate change? Land use change?
Line 194: what kind of variable are “land use” and “climate”?
Line 195: what kind of variable is „microbiology”? Analysis analysis?
Line 196: ] ?
Line 196: linear relationships between what kind of data?
Line 201: vegan package reference?
You should have investigated the correlations among explanatory variables because there may have been many collinearities. (As we can see it on plot of RDA.)
Results
Line 204: these terms with „changes” are really confusing. In Figure 1, Land use type is used. Why don’t you also use this term elsewhere?
Line 205: the factor climate had significant effects on…
Lines 205, 207 and Table 1: G- G+??? What is that? In lines 179-180 you wrote that G- is Gram negative, but not Gram negative bacteria, so you should write down G- bacteria, or you should determine the abbreviation more clearly. Also in Table 1: Gram negative, gram positive? What is that?
Line 205: anaerobe what? Anaerob conditions? (I know that they are anaerob bacteria but this is not enough, it is careless writing.)
Line 206: eukaryotes? This is the first time you mention them. These are not fungi? Then what? Algae? Fungi are also eukaryotes… This is confusing.
Line 207: land use change…
205-251: We can see levels of significance in Tables 1-2-3, but you shouldn’t write it down twice (also in the main text). I can see that these are (most of the time) the original p-values. You should rather write the accurate p-values into the tables instead of the main text. The tables should include the exact p-values and leave them from the main text.
Table 1/Table 2: It is strange that you use asterics for levels of significance, and also you write down whether it is lower than 0.01, 0.001 etc. We do not use it in this way. You should write the ORIGINAL p-values, and then you can take * or ** or *** beside the original p-value. In the actual way in your manuscript, this is redundant, why do you use *** and also write down <0.001?
Line 210: Fungi or fungi?
Table 2: F-values…
Lines 224, 246: It is also very confusing that you use in the text „HTHH climate change” (which is quite strange in itself) but in Figures 1-3 you use the names of areas (which makes much more sense). It should be uniform. Why do I have to learn which is HTHH, Huanan or Youyi?
Lines 224, 226: My main problem again: „increased”, „changed significantly”. If you compare different areas, and you can find differences between them, it means that there are differences between them. It does not mean changes!
Lines 247, 251: life forms OF Collembola
Figure 3 is missing. The caption is there but the plot is missing.
Line 281, 283: species name with italic font style
Line 281: I really missed from Table S2 at least the mean frequency of the species. And in Figure 5 you cannot read all species names, they are on each other, unreadable. Figure 5 should also be corrected somehow.
Figure 5: It seems like the most microorganism variables [mainly bacteria (G+, G-, anaerobe) and fungi] correlated to each other. I suggest to calculate variance inflation factor (VIF) for the variables and then leave some of the correlated explanatory variables. You may be able to make a more accurate model then.
Figure 5: “Land use” is a factor. How come that it appears with an arrow in the plot? We should have seen the factor levels on the plot.
Line 292: “Land use practices” A third name. I prefer this. Much better that “land use change”.
Line 293: Gram Negative again… Gram negative bacteria… Eukaryote??? Fungi are also eukaryotes…
Discussion
Line 302: My main problem with the conception of this manuscript is that it does not study climate change! It compared different areas with slightly different climates. It is not about climate change!
Line 305: You refer to studies (28, 67, 68): these studies are climate manipulation studies. These are real climate change studies. But your study is not a climate change study!
Line 317: NOT land use change! Just “Land use practice”. No change!
Lines 331-334: These conclusions make no sense in light of the fact that the study is not a manipulative climate change study at all.
Line 361: not increase! Difference! It is, again, a spatial difference, not a temporal change!
Line 374: the reference #75 contains a study about marshlands, not drylands. Whatever, this study is really an experiment about land use change… not like yours.
Line 384: what is an upland soil? A brand new term in the last section. Strange.
Author Response
Reviewer: 3
Comments to the Author
The manuscript compares the Collembola assemblages of three areas with three types of crop plant. The study has interesting and useful results, however, the authors do not interpret their own results properly. At first I thought they were just using the concepts incorrectly, but later I realized they were also interpreting them incorrectly. My main problem is this: climate change and land use change? The study did not investigate changes but rather differences! Change is the term, if something becomes different in time, but you investigated static areas with given climates ONCE. If you want to investigate changes, you should sample your objects several times and then you can see whether it will change or not. This is a serious problem, and without it being corrected, this manuscript cannot be published in my opinion.
Response: Thank you for your valuable comments and suggestions on our research. We highly appreciate your feedback, which we have carefully considered and will incorporate into the revised manuscript. On reflection we realised we had got the concept wrong and we have now changed the text from climate and land use change to different climate conditions and land use practices.
Title:
Comments 1: This study does not investigate climate change. I suggest to reword the title. This is now misleading.
Response 2: Agree, thank you very much for this suggestion. we modified the title: Euedaphic Rather than Hemiedaphic or Epedaphic Collembola are More Sensitive to Different Climate Conditions in the Black Soil Region of Northeast China.
Abstract:
Comments 2: Line 22: You compared three different areas with slightly different climates. What kind of change is it?
Response 2: Agree, thank you very much for this suggestion.We have modified climate change and replaced it with different climate conditions and change the sentence: Here, we selected three climatic areas from high to low latitudes in the black soil region of Northeast, with three variations land use practices (soybean, maize, and rice) sampled in each area, please see lines 40-42. We have attached the mean MAT and MAP between sampling sites for each area in section 2.1 Study area (lines 133-153), where the mean MAT is 2.61°C and MAP is 556 mm in Fujin (lines 133-138); the mean MAT is 3.23°C and MAP is 567 mm in Huanan (lines 139-145); Youyi's mean MAT was 3.58°C and MAP was 532 mm (lines 146-151). We wrote in the introduction (lines 104-108) section that previous studies have found an increasing trend in temperature of 0.31°C/l0a in the black soil area of Northeast, whereas precipitation is irregularly increasing. In our sampling sites, compared with Fujin, Huanan has higher temperature and higher humidity with MAT increase of 0.62°C and MAP increase of 11 mm, we modelled 20 years of temperature change using these two regions; compared with Fujin, Youyi has higher temperature and lower humidity with MAT increase of 0.97°C and MAP decrease of 23 mm, we modelled 30 years of temperature change using these two regions.
Comments 3: Line 26: “higher temperature and higher humidity (HTHH) climate change” This is climate difference. This is not change.
Response 3: Agree. We recognise that is a climate difference and change the sentence: We found that higher temperature and higher humidity and land use practices from rice to soybean and maize are associated with higher Collembola density and species rich-ness. please see lines 42-44.
Comments 4: Line 27: “land use change from rice to soybean and maize” It would be a change, if the crop land would have been previously rice then it would have changed into soybean. But in your study it is not the case. This word change and also this sentence make no sense. This “land use change” cannot “increase” the density of Collembola. I tell you what you detected. You detected different density and diversity values between the land use practices. And the same problem is with “climate change” in the WHOLE MANUSCRIPT. You need to rethink what you tested, what results you got, and what it all means.
Response 4: Agree, thank you very much for this suggestion. We recognise that you have made a very valid point, and we also recognise that our experiment compares differences rather than changes, so we have changed land use change to land use practices difference.
Introduction:
Comments 5: The Introduction is misleading. The reader thinks that the topic and the study will be about a real climate change and land-use change experiment. But all the investigation is not about such an experiment.
Response 5: Agree, thank you very much for this suggestion. We recognise that you have made a very valid point, and we also recognise that our experiment is comparing differences rather than changes. We changed the sentences in the introduction about climate and land use change.
Comments 6: Line 81: forests and grasslands.
Response 6: Agree. We change to “forests and grasslands”, please see line 99.
Comments 7: Lines 95-108: I suggest to rewrite all the hypotheses because you did not investigate climate and land use change…
Response 7: Agree, thank you very much for this suggestion. We rewrite all the hypotheses: In light of a clear lack of research on the subject, the study investigates the differences on Collembola communities under varying climate conditions and land use practices in the black soil region of Northeast China.
Firstly, the differences in climate conditions and land use practices result in different soil environments and feeding resources, such as temperature, humidity and soil microorganisms [47-49]. Therefore, we hypothesize (H1) that Collembola density and species richness respond differently to different climate conditions and land use types.
Secondly, euedaphic Collembola, which have limited mobility when compared to hemiedaphic and epedaphic Collembola [31], we hypothesize (H2) that euedaphic Col-lembola are more sensitive to different climate conditions and land use practices.
Thirdly, feeding resources such as SOMs are relatively high in the black soil region and this could not be a limited factors to Collembola community [50], so we hypothe-size (H3) that the soil environment conditions (e.g. whether it is flooded or not, tem-perature, etc.), will have a greater influence on the Collembola than the feeding re-source (microorganisms).
Please see lines 111-126.
Materials and methods:
Comments 8: Line 115: what does CK stand for?
Response 8: Agree, we chose Fujin, with the lowest temperature and humidity, as the control group, the “CK” represents the control group.
Comments 9: Lines 126, 133: It is very strange that you call „climate change” the geographical differences between climates. Climate change means that the climate changes IN TIME, not SPACE! I suggest not to call it climate change, maybe „difference in climate”.
Response 9: Agree. We modified the sentence and removed the climate change. Please see lines 144 and 150.
Comments 10: A map about the study sites would tell more than Table S1.
Response 10: Agree. We have added a geographic map of the sampling locations Table S2.
Comments 11: Lines 135-136: Three “static” climatic area will be compared. Where is the climate change? Did these areas had lower temperatures and higher precipitations some years ago? Did you also investigate the Collembola some years ago on these areas? Then this study is not about climate change…
Response 11: Agree. We modified the sentence: By selecting these three distinct climatic areas, we aimed to capture the variations in climate conditions present in the black soil area of Northeast China. Please see lines 152-153.
Comments 12: Lines 138-141: Three types of crop plants. Were these areas had previously different crop plants e.g. in 2022 or in 2021? Did you investigate the Collembola before on these areas? Then where is “land use CHANGE”?
Response 12: Agree, thank you very much for this suggestion. We recognise that you have made a very valid point, and we also recognise that our experiment compares differences rather than changes. We have changed the sentence in the text about land use change.
Comments 13: Line 144: what are the amounts and timings of pesticide uses?
Response 13: Agree. We rewrite the sentence: as well as Pesticide use is also consistent, with the use of herbicides (ethopropylamine, atrazine) (1500-3000 ml/ha) and insecticides (750-1500 ml/ha) and these pesticides are used in the spring sowing and summer crop growth periods, please see lines 161-164.
Comments 14: Line 190: Climate change? Land use change?
Response 14: Agree. We rewrite the sentence: Therefore, we used two-way ANOVA to analyze differences in all response variables under varying climate conditions and land use practices. Please lines 208-210.
Comments 15: Line 194: what kind of variable are “land use” and “climate”?
Response 15: Agree. We modified land use practices and climate conditions. The variable of land use practices represent the differences between maize, soybean and rice fields; the variable of climate conditions represent Climate differences in the three regions (mainly MAT and MAP).
Comments 16: Line 195: what kind of variable is „microbiology”? Analysis analysis?
Response 16: Agree. We modified to “microorganisms” and delete the analysis. Please see line 214.
Comments 17: Line 196: ] ?
Response 17: Agree. We delete the “]”.
Comments 18: Line 196: linear relationships between what kind of data?
Response 18: Agree. We rewrite the sentence: We checked for linear relationships in the data sets (Euclidean metric; prerequisite for this method) by detrended correspondence analyses (DCA) and identifying the respec-tive longest gradient. As these were always lower than 3, the use of linear methods was considered appropriate. Please see lines 216-219.
Comments 19: Line 201: vegan package reference?
Response 19: Agree. We add vegan package reference. Please see line 221.
Comments 20: You should have investigated the correlations among explanatory variables because there may have been many collinearities. (As we can see it on plot of RDA.)
Response 20: Agree, thank you very much for this suggestion. In this experiment, the most dominant influencing factor is land use practice difference, while other factors have much less influence than land use practice difference, and we wrote in section 3.1 that microorganisms under land use practice difference are This also suggests that land use practice difference and climate difference can explain these effects.
Results:
Comments 21: Line 204: these terms with „changes” are really confusing. In Figure 1, Land use type is used. Why don’t you also use this term elsewhere?
Response 21: Agree, thank you very much for this suggestion. We recognise that you have made a very valid point, and we also recognise that our experiment compares differences rather than changes. We have changed the sentence in the text about land use change.
Comments 22: Line 205: the factor climate had significant effects on…
Response 22: Agree. We rewrite the sentence: There are significant differences in AMF, Gram Negative, Anaerobe, and Actinomycetes, while there were no significant differences in Eukaryote, Fungi, and Gram Positive under climate conditions difference. Please see lines 224-226.
Comments 23: Lines 205, 207 and Table 1: G- G+??? What is that? In lines 179-180 you wrote that G- is Gram negative, but not Gram negative bacteria, so you should write down G- bacteria, or you should determine the abbreviation more clearly. Also in Table 1: Gram negative, gram positive? What is that?
Response 23: Agree, thank you very much for this suggestion. We delete the abbreviation and change to Actinomycetes, Anaerobe, Gram Positive and Gram Negative.
Comments 24: Line 205: anaerobe what? Anaerob conditions? (I know that they are anaerob bacteria but this is not enough, it is careless writing.)
Response 24: Agree. We delete the abbreviation and change to Actinomycetes, Anaerobe, Gram Positive and Gram Negative.
Comments 25: Line 206: eukaryotes? This is the first time you mention them. These are not fungi? Then what? Algae? Fungi are also eukaryotes… This is confusing.
Response 25: Agree. The Eukaryotes refers to Fungi and Protists.
Comments 26: Line 207: land use change…
Response 26: Agree. We have changed the sentence in the text about land use change.
Comments 27: 205-251: We can see levels of significance in Tables 1-2-3, but you shouldn’t write it down twice (also in the main text). I can see that these are (most of the time) the original p-values. You should rather write the accurate p-values into the tables instead of the main text. The tables should include the exact p-values and leave them from the main text.
Table 1/Table 2: It is strange that you use asterics for levels of significance, and also you write down whether it is lower than 0.01, 0.001 etc. We do not use it in this way. You should write the ORIGINAL p-values, and then you can take * or ** or *** beside the original p-value. In the actual way in your manuscript, this is redundant, why do you use *** and also write down <0.001?
Response 27: Agree. We remove the p-values of the main text, and we write the accurate p-values into the tables.
Comments 28: Line 210: Fungi or fungi?
Response 28: Agree. It’s Fungi.
Comments 29: Table 2: F-values…
Response 29: Agree. We modified with “F-values”. Please see lines 231, 248 and 272.
Comments 30: Lines 224, 246: It is also very confusing that you use in the text „HTHH climate change” (which is quite strange in itself) but in Figures 1-3 you use the names of areas (which makes much more sense). It should be uniform. Why do I have to learn which is HTHH, Huanan or Youyi?
Response 30: Agree. We remove the abbreviation of HTHH and we use the higher temperature and higher humidity.
Comments 31: Lines 224, 226: My main problem again: „increased”, „changed significantly”. If you compare different areas, and you can find differences between them, it means that there are differences between them. It does not mean changes!
Response 31: Agree. We rewrite the sentence: On the other hand, under higher temperature and higher humidity, the total Collembola density and species richness much higher; while under higher temperature and lower humidity, total Collembola density and species richness did not differ significantly. Please see lines 242-245.
Comments 32: Lines 247, 251: life forms OF Collembola.
Response 32: Agree. We change to “life forms of Collembola”. Please see lines 266 and 271.
Comments 33: Figure 3 is missing. The caption is there but the plot is missing.
Response 33: Agree. We rechecked the paper and added Figure 3.
Comments 34: Line 281, 283: species name with italic font style.
Response 34: Agree. We modified with “Thalassaphorura macrospinata” and “Proisotoma minuta”. Please see lines 301 and 304.
Comments 35: Line 281: I really missed from Table S2 at least the mean frequency of the species. And in Figure 5 you cannot read all species names, they are on each other, unreadable. Figure 5 should also be corrected somehow.
Response 35: Agree. We add the relative abundance of Collembola species in Table S3.
Comments 36: Figure 5: It seems like the most microorganism variables [mainly bacteria (G+, G-, anaerobe) and fungi] correlated to each other. I suggest to calculate variance inflation factor (VIF) for the variables and then leave some of the correlated explanatory variables. You may be able to make a more accurate model then.
Response 36: Agree, thank you very much for this suggestion. Environmental factors are a greater influence than feeding resources, and Collembola ingests differently, so we selected all microorganisms.
Comments 37: Figure 5: “Land use” is a factor. How come that it appears with an arrow in the plot? We should have seen the factor levels on the plot.
Response 37: Agree, we made a new Figure 5, and we change to “land use practices”.
Comments 38: Line 292: “Land use practices” A third name. I prefer this. Much better that “land use change”.
Response 38: Agree, thank you very much for this suggestion. We use the “land use practices”.
Comments 39: Line 293: Gram Negative again… Gram negative bacteria… Eukaryote??? Fungi are also eukaryotes…
Response 39: Agree. We change to Actinomycetes, Anaerobe, Gram Positive and Gram Negative.
Discussion:
Comments 40: Line 302: My main problem with the conception of this manuscript is that it does not study climate change! It compared different areas with slightly different climates. It is not about climate change!
Response 40: Agree, thank you very much for this suggestion. We recognise that you have made a very valid point, and we also recognise that our experiment is comparing differences rather than changes, so we have changed the one about changes to differences.
Comments 41: Line 305: You refer to studies (28, 67, 68): these studies are climate manipulation studies. These are real climate change studies. But your study is not a climate change study!
Response 41: Agree, thank you very much for this suggestion. We recognise that you have made a very valid point, and we also recognise that our experiment is comparing differences rather than changes, so we have changed the one about changes to differences.
Comments 42: Line 317: NOT land use change! Just “Land use practice”. No change!
Response 42: Agree, we change “land use change” to “land use practices”.
Comments 43: Lines 331-334: These conclusions make no sense in light of the fact that the study is not a manipulative climate change study at all.
Response 43: Agree. We rewrite the sentence: While different climate conditions do affect Collembola communities, its impact does not mitigate the effects of difference in land use practices. Consequently, it can be inferred that the influence of difference in land use practices on Collembola communities remains unaffected by varying climate conditions. Please see lines 352-355.
Comments 44: Line 361: not increase! Difference! It is, again, a spatial difference, not a temporal change!
Response 44: Agree. We rewrite the sentence: Our study revealed that higher MAP and MAT resulted in a higher species richness of hemiedaphic Collembola. Please see lines 383-384.
Comments 45: Line 374: the reference #75 contains a study about marshlands, not drylands. Whatever, this study is really an experiment about land use change… not like yours.
Response 45: Agree. The reference #75 are not appropriate here, so we used a more appropriate reference: 73. Saifutdinov, R.A.; Sabirov, R.M.; Zaitsev, A.S. Springtail (Hexapoda: Collembola) functional group composition varies be-tween different biotopes in Russian rice growing systems. European Journal of Soil Biology. 2020, 99, 103208.
Comments 46: Line 384: what is an upland soil? A brand new term in the last section. Strange.
Response 46: Agree, we change “upland soil” to “drylands”, please see line 407.

Round 2
Reviewer 3 Report
Comments and Suggestions for Authors
I thank the authors that they revised their manuscript according to my comments. Most of them are acceptable, however, I have some new comments to them.
Comment 1
Accepted
Comment 2
Accepted
Comment 3
Accepted
Comment 4
Accepted
Comment 5
Accepted
Comment 6
Accepted
Comment 7
These hypotheses are good. Still two little things:
The sentence about the second hypothesis is strange. There is no verb for the first mention of euedaphic Collembola.
In addition, you should describe SOM at the first mention.
Comment 8
Accepted. I understood it I was just wondering what CK literally means. I thought that C is for control, but what does K mean? However, it is not an important point.
Comment 9
Accepted
Comment 10
Thank you for the map. It looks nice and is almost good. It could be better if the map focused on the research area not on the whole province. With higher resolution it would be perfect.
Comment 11
Accepted
Comment 12
Accepted
Comment 13
Accepted
Comment 14
Accepted
Comment 15
Accepted
Comment 16
I would add to the variables microorganisms that “and the measured PLFA values of the microorganisms”, because we know that land use practices is a three-level factor, climate is also a three-level factor, but what is microorganisms? Be more accurate!
Comment 17:
No, you did not delete the “]”.
Comment 18
Accepted
Comment 19
Accepted
Comment 20
Accepted
Comment 21
Accepted
Comment 22
Accepted
Comment 23
My problem was that you used imprecisely these terms "Gram positive bacteria", "Gram negative bacteria" etc.
Or you should have described all your terms in the part: lines 197-202. E.g. Gram positive bacteria PLFA ("Gram positive") [...].
Besides, you left anaerob bacteria ("Anaerobe") from your list.
Comment 24
You also deleted Anaerobs from the M&M parts.
Comment 25
It is not acceptable. You did not describe this information in the MS. Eukaryotes were also not mentioned in the part of lines 197-202. And how come that there is a separate group for Fungi and Protist when you have another group for AMF? This grouping is strange.
Comment 26
Where? In which line?
Comment 27
It is okay. But leave that "are presented in Table 1".
Comment 28
But see Table 1 caption: fungi.
Comment 29
Accepted
Comment 30
Accepted
Comment 31
"were" much higher, right?
Comment 32
Accepted
Comment 33
Accepted
Comment 34
Accepted
Comment 35
I couldn't find it in Table S3 but I found it in Figure S3. Thank you!
Comment 36
Accepted
Comment 37
Leave it.
Comment 38
Accepted
Comment 39
And what about the "Eukaryote" in Table 1, Table 4, and Figure 5?!
Comment 40
I see that you changed all the climate change terms into differences. However, I would have thought that this would mean that the discussion section would have to be completely rewritten, as it is a completely new concept. But then I guess not. Even the references could have remained the same. Interesting.
Comment 41
Accepted
Comment 42
Accepted
Comment 43
Accepted
Comment 44
Accepted
Comment 45
Accepted
Comment 46
Accepted
Author Response
Comments to the Author
I thank the authors that they revised their manuscript according to my comments. Most of them are acceptable, however, I have some new comments to them.
Response: Thank you for your valuable comments and suggestions on our research. We highly appreciate your feedback, which we have carefully considered and will incorporate into the revised manuscript.
Comments 1: These hypotheses are good. Still two little things:
The sentence about the second hypothesis is strange. There is no verb for the first mention of euedaphic Collembola.
In addition, you should describe SOM at the first mention.
Response: Thanks for you suggestion. We modified the sentence: Secondly, in contrast to hemiedaphic and epedaphic forms, euedaphic Collembola demonstrate substantially restricted mobility patterns. Please see lines 199-120.
We add the full name of the SOM, which is soil organic matter. Please see line 123.
Comments 2: Accepted. I understood it I was just wondering what CK literally means. I thought that C is for control, but what does K mean? However, it is not an important point.
Response: Thanks for you suggestion. Generally speaking. CK is an abbreviation for Control check, and defaults to CK. So we use the CK.
Comments 3: Thank you for the map. It looks nice and is almost good. It could be better if the map focused on the research area not on the whole province. With higher resolution it would be perfect.
Response: Thanks for you suggestion. We made a higher resolution map of sampling points, please see Figure S2.
Comments 4: I would add to the variables microorganisms that “and the measured PLFA values of the microorganisms”, because we know that land use practices is a three-level factor, climate is also a three-level factor, but what is microorganisms? Be more accurate!
Response: Thanks for you suggestion. We modified to “the measured PLFA values of the microorganisms”. Please see line 218.
Comments 4: No, you did not delete the “]”.
Response: Thanks for you suggestion. We delete the “]”, please line 219.
Comments 5: My problem was that you used imprecisely these terms "Gram positive bacteria", "Gram negative bacteria" etc.
Or you should have described all your terms in the part: lines 197-202. E.g. Gram positive bacteria PLFA ("Gram positive") [...].
Besides, you left anaerob bacteria ("Anaerobe") from your list.
Response: Thanks for you suggestion. We reviewed the literature again regarding the proper way to write about Gram-positive bacteria and we modified to Gram-positive bacteria (G+) and Gram-negative bacteria (G-) in lines 198-199, the abbreviation of G+ and G- was used throughout the text.
We have included anaerobe and eukaryote related markers in PLFA, please see lines 201-205.
Comments 6: You also deleted Anaerobs from the M&M parts.
Response: Thanks for you suggestion. We have included anaerobe related markers in PLFA, please see lines 201-202.
Comments 6: It is not acceptable. You did not describe this information in the MS. Eukaryotes were also not mentioned in the part of lines 197-202. And how come that there is a separate group for Fungi and Protist when you have another group for AMF? This grouping is strange.
Response: Thanks for you suggestion. The fungi contain AMF, while the eukaryotes contain fungi and other microorganisms. We have included eukaryote related markers in PLFA, please see lines 203-205.
Comments 7: Where? In which line?
Response: Thanks for you suggestion. We modified to different land use practices, please see line 231.
Comments 8: It is okay. But leave that "are presented in Table 1".
Response: Thanks for you suggestion. We leave that "are presented in Table 1".
Comments 9: But see Table 1 caption: fungi.
Response: Thanks for you suggestion. We rechecked the case issue and used lowercase uniformly in the article, e.g. fungi, actinomycete, anaerobe and eukaryote.
Comments 10: "were" much higher, right?
Response: Thanks for you suggestion. We modified to “were much higher”, please see line 245.
Comments 11: And what about the "Eukaryote" in Table 1, Table 4, and Figure 5?!
Response: Thanks for you suggestion. The eukaryotes contain fungi and other microorganisms. We have included eukaryote related markers in PLFA, please see lines 203-205.
Comments 12: I see that you changed all the climate change terms into differences. However, I would have thought that this would mean that the discussion section would have to be completely rewritten, as it is a completely new concept. But then I guess not. Even the references could have remained the same. Interesting.
Response: Thanks for you suggestion. We rechecked the references in the article again and there are indeed some references to climate and land use change that are not quite suitable. So we researched these references and replaced them with references on climate conditions and land use practices.